# LEARNING EQUI-ANGULAR REPRESENTATIONS FOR ONLINE CONTINUAL LEARNING

## ABSTRACT

Online continual learning suffers from an underfitted solution for prompt model update due to the constraint of single-epoch learning. We confront this challenge by proposing an efficient online continual learning method with the notion of neural collapse. In particular, we induce neural collapse to form a simplex equiangular tight frame (ETF) structure in the representation space so that the learned model with single epoch can better fit the streamed data by proposing preparatory data training and residual correction in the representation space. With an extensive set of empirical validations using CIFAR10/100, TinyImageNet, and ImageNet-200, we show that our proposed method outperforms state-of-the-art methods by a noticeable margin in various online continual learning scenarios, including Disjoint and Gaussian scheduled setups.

## 1 INTRODUCTION

A burgeoning surge of interest in continual learning (CL) involves training the model using continuous data streams. Predominantly, CL research has focused on the *offline* CL setup that assumes the model can be trained in multiple epochs for the current task (Rebuffi et al., 2017; Chaudhry et al., 2018; Wu et al., 2019). However, it requires substantial storage capacity to store all the data of the current task for training multiple epochs. Recently, there has been significant interest in *online* CL as a more realistic set-up with less computational overhead of allowing a single pass through the data stream (Aljundi et al., 2019; Koh et al., 2022; Cai et al., 2021). As we are supposed to update the model for every data batch in online CL, even if the overall distribution of a dataset is balanced, the temporal distribution at each intermediate time point in the data stream is likely imbalanced.

Imbalanced data distributions would cause several problems, such as bias towards the major classes (Zhao et al., 2021; Kang et al., 2021) and the hindrance to generalization (Wu, 2023). Recently, *minority collapse* (Fang et al., 2021), the phenomenon in which angles between classifier vectors for minor classes become narrow, has been proposed as a fundamental issue in training with imbalanced data, making the classification of minor classes considerably more challenging. On the contrary, for balanced datasets, it was proven that classifier vectors and the last layer activations for all classes converge into an optimal geometric structure, named the simplex *equiangular tight frame* (ETF) structure, where all pairwise angles between classes are equal and maximally widened when using cross-entropy (CE) (Ji et al., 2021; Lu & Steinerberger, 2020; Zhu et al., 2021; Wojtowytsch et al., 2020) or mean squared error (MSE) (Zhou et al., 2022b; Mixon et al., 2020; Rangamani & Banburski-Fahey, 2022; Tirer & Bruna, 2022) loss. This phenomenon is called *neural collapse* (Papyan et al., 2020). Although neural collapse naturally occurs only in balanced training, several recent studies attempted to induce neural collapse in imbalanced training to address the minority collapse problem using a fixed ETF classifier (Yang et al., 2022; Zhong et al., 2023). Very recently, research has also been extended to induce neural collapse in offline CL scenarios (Yang et al., 2023a).

However, in online CL, there are a number of challenges to inducing neural collapse. The prerequisite for neural collapse is reaching the *terminal phase of training* (TPT) by sufficient training (Papyan et al., 2020). In offline CL, the model can reach the TPT phase for each task by multi-epoch training. In contrast, it is challenging to reach TPT in online CL due to its limitation of allowing only a single-pass training except for data stored in episodic memory, and the data stored within the memory keeps changing as new samples come in. As shown in Fig. 1, the offline CL quickly reaches TPT shortly after adding novel task data, while online CL (vanilla ETF) does not.

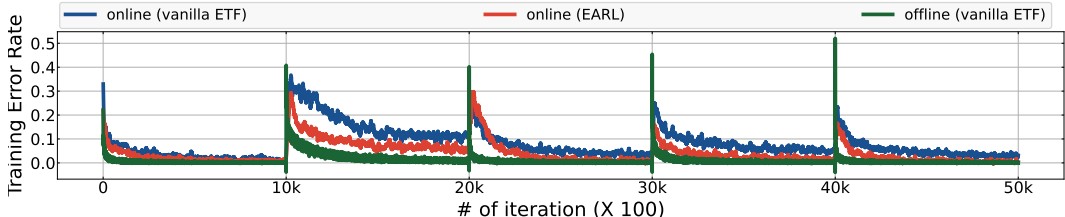

Figure 1: Comparison of training error rates between online CL and offline CL in the CIFAR-10 disjoint setup, where two novel classes are added every 10k samples. Vanilla ETF refers to a method that both preparatory data training and residual correction are removed from the proposed EARL.

Recently, the importance of anytime inference in online CL has been emphasized (Pellegrini et al., 2020; Koh et al., 2022; Caccia et al., 2022; Ghunaim et al., 2023), since a model should be able to be used for inference not only at the end of a task, but also at any point during training to be practical for real-world applications. Hence, not only reaching TPT but also achieving faster convergence is essential when using neural collapse in online CL.

However, the phenomenon in which the features of the new class become biased towards the features of the existing classes hinders the fast convergence of the last-layer features into the ETF structure. When features of old and novel classes overlap and are trained with the same objective, the well-clustered features of the old classes get perturbed, leading to the destruction of the ETF structure formed by the features of old classes.

To address this issue, we propose using preparatory data in training, which are obtained by applying hard transforms to data of existing classes, to distinguish old class and novel class when a novel class arrives. This promotes fast and stable convergence into the ETF structure. However, despite these efforts, the continuous stream of new samples prevents them from reaching the TPT and fully converging to the ETF structure. To address this, we propose to store the residuals between the target ETF classifier and the features during training, and during inference, we correct the inference output using the stored residual to compensate for insufficient convergence in training. By acceleration of convergence by preparatory data training and additional correction using residual. We name our method **Equi-Angular Representation Learning** (**EARL**). We demonstrate the effectiveness of our framework on CIFAR-10, CIFAR-100, TinyImageNet and ImageNet-200. To be specific, our framework outperforms various CL methods by a significant margin (+4.0% gain of $A_{\text{auc}}$ in ImageNet-200).

**Contributions.** We summarize our contributions as follows:

- Proposing to induce neural collapse for online continual learning.
- Proposing 'preparatory data training' to address the 'bias problem' that the new classes are biased toward the existing classes, promoting faster induction of neural collapse.
- Proposing 'residual correction' scheme to compensate for not fully reaching neural collapse at inference to further improve anytime inference accuracy.

## 2 RELATED WORK

**Continual learning methods.** Various continual learning methods are being researched to prevent forgetting past tasks, broadly categorized into replay, parameter isolation, and regularization. Replay methods involve storing a small portion of data from previous tasks in episodic memory (Hayes et al., 2020; Aljundi et al., 2019; Koh et al., 2022; Bang et al., 2021; Yoon et al., 2021) or storing a generative model trained on data from previous tasks (Shin et al., 2017; Pomponi et al., 2023). By replaying the samples stored in episodic memory or generated from the stored generative model, the model prevents forgetting past tasks during subsequent learning of novel tasks. Furthermore, Boschini et al. (2022); Buzzega et al. (2020); Li & Hoiem (2017); Wu et al. (2019) used replay samples to distill information about past tasks.

Regularization methods (Kirkpatrick et al., 2017; Lesort et al., 2019) apply a penalty to important model parameters that change during the process of learning new tasks, allowing the model to retain information about previous tasks. Parameter isolation methods (Zhou et al., 2022a; Rusu et al.,

2016; Cheung et al., 2019) expand the network by allocating specific layers for each task. This enables the network to store information about individual tasks and preserves their knowledge without forgetting.

Data are often assumed to be divided into tasks with explicit task boundaries, where the distribution changes rapidly (Cai et al., 2021), and many methods (Wu et al., 2019; Ye & Bors, 2021; Bang et al., 2021; Sun et al., 2022) use the information about the task boundary during training. However, in practical real-world scenarios, the availability of task boundaries is a rarity. Consequently, several task-agnostic approaches have been introduced (Aljundi et al., 2018; Koh et al., 2022; Ye & Bors, 2022b), while the majority of methods still incorporate task boundaries during the training process (Kirkpatrick et al., 2017; Yoon et al., 2021; Ye & Bors, 2022a; Zhou et al., 2022a; Boschini et al., 2022). Our proposed method also refrains from using task identity information during training, aiming to simulate a more realistic scenario where task boundaries are not provided.

**Anytime inference in online continual learning.** In online CL, new data arrive continuously in a stream rather than a large chunk (*e.g.*, task unit). Several previous works (Bang et al., 2021; Kim et al., 2021) assumed to wait until a large chunk of new data accumulates before training the model. However, this approach has a limitation in that it only exhibits good inference performance at the point of model update using the large chunk, as no learning occurs during the waiting period (Koh et al., 2022; Caccia et al., 2022).

In contrast, since inference queries to the model can occur at any time, not just immediately after a model update, there is active research on the importance of anytime inference (Pellegrini et al., 2020; Koh et al., 2022; Doan et al., 2022; Ghunaim et al., 2023; Banerjee et al., 2023) recently. Consequently, we focus not only on $A_{last}$, which corresponds to the point when learning has finished for all data, but also on $A_{\text{AUC}}$, which measures the 'anytime' accuracy (Koh et al., 2022).

**Neural collapse.** Neural collapse is a phenomenon in which the activations of the last layer and the classifier vectors form a simplex *equiangular tight frame* (ETF) structure at the terminal phase of training (TPT) in a balanced dataset. (Papyan et al., 2020). Neural collapse has been demonstrated as the global optimum of balanced training using CE (Ji et al., 2021; Lu & Steinerberger, 2020; Zhu et al., 2021; Wojtowytsch et al., 2020) and MSE (Zhou et al., 2022b; Mixon et al., 2020; Rangamani & Banburski-Fahey, 2022; Tirer & Bruna, 2022) loss functions, within a simplified model focused solely on last-layer optimization. Inducing neural collapse in imbalanced datasets poses challenges due to minority collapse (Fang et al., 2021) where minor classes are not well distinguished.

However, using a fixed ETF classifier, it is empirically and theoretically shown that neural collapse was induced even in imbalanced datasets (Yang et al., 2022). Continual learning also needs to address imbalanced data since there is imbalance in data between novel classes and existing classes. Therefore, in offline CL, NC-FSCIL (Yang et al., 2023a) used a fixed ETF classifier to induce neural collapse. Meanwhile, online CL often fails to induce neural collapse compared to offline CL, *e.g.*, FSCIL, since it restricts sufficient training in multiple epochs, causing the failure to reach TPT.

## 3 PRELIMINARIES

We provide background knowledge about neural collapse and equiangular tight frame (ETF) classifier here, and describe a formal problem statement for the online continual learning in Sec. A.1 in Appendix for the sake of space.

### 3.1 NEURAL COLLAPSE

Neural collapse (Papyan et al., 2020) is a phenomenon of the penultimate features after convergence of training on a balanced dataset. Specifically, if the neural collapse (NC) occurs, the collection of K classifier vectors $\mathbf{W}_{\text{ETF}} = [w_1, w_2, ..., w_K] \in \mathbb{R}^{d \times K}$ forms a simplex equiangular tight frame (ETF), which satisfies:

$$w_i^T w_j = \begin{cases} 1, & i = j \\ -\frac{1}{K-1}, & i \neq j \end{cases} \quad, \quad \forall i, j \in [1, ..., K], \tag{1}$$

and the penultimate feature of a training sample collapses into an ETF vector $w_i$. Here, we brief the notion of ETF and necessary metrics to measure how much the model forms the ETF structure. We discuss them in detail in the Appendix A.11 for the sake of space.

**(NC1) Collapse of variability**: The last layer activation $h_{k,i}$ of sample $i$ in class $k$ collapses to $\mu_k = \frac{1}{n_k} \sum_{i=1}^{n_k} h_{k,i}$ for $\forall k \in [1, ..., K]$ where $n_k$ is the number of samples for class $k$, *i.e.*, $\Sigma_W \rightarrow 0$, where $\Sigma_W = \frac{1}{K} \sum_{k=1}^{K} \frac{1}{n_k} (\sum_{i=1}^{n_k} (h_{k,i} - \mu_k))$ represents within-class covariance.

**(NC2) Convergence to simplex equiangular tight frame (ETF)**: Class means $\mu_k (k \in [1, ..., K])$ centered by the global mean $\mu_G = \frac{1}{K} \sum_{k=1}^{K} \mu_k$ converge to vertices of a simplex ETF structure, *i.e.*, $\mathbf{MM}^T = \frac{1}{K-1}(K\boldsymbol{I}_k - \mathbf{1}_K \mathbf{1}_K^T)$

**(NC3) Convergence to self-duality**: Classifier $\mathbf{W}$ converges to $\mathbf{M}$ formed by recentered feature mean $m_k = \frac{\mu_k - \mu_G}{\|\mu_k - \mu_G\|^2}$ *i.e.*, $\frac{\mathbf{M}}{\|\mathbf{M}\|_F} = \frac{\mathbf{W}}{\|\mathbf{W}\|_F}$

**(NC4) Simplification to nearest class center (NCC)**: Class predicted by the last layer activation corresponds to the nearest class center, *i.e.*, $\arg\max_k \langle h_{k,i}, w_k \rangle = \arg\min_k \|h_{k,i}, \mu_k\|$

### 3.2 Equiangular Tight Frame (ETF) Classifier

Inspired by neural collapse as described in Sec. 3.1, a fixed ETF classifier has been utilized for inducing neural collapse in imbalanced datasets (Zhu et al., 2021; Yang et al., 2022; 2023a). Here, the classifier is initialized by the ETF structure $\mathbf{W}_{\text{ETF}}$ at the beginning of training and fixed during training to induce the penultimate feature $f(x)$ to converge to the ideal balanced scenario. For training $f(x)$, it is only required to attract $f(x)$ to the corresponding classifier vector for convergence since the classifier is fixed during training. Therefore, following Yang et al. (2022), we use the *dot regression* (DR) loss as a training objective, as it shows to outperform CE loss when using a fixed ETF classifier in imbalanced datasets (Yang et al., 2022). We write our DR loss as:[1]

$$\mathcal{L}_{\text{DR}}(\hat{f}(x), y; \mathbf{W}_{\text{ETF}}) = \frac{1}{2} \Big( \mathbf{w}_y \hat{f}(x) - 1 \Big)^2, \quad \hat{f}(x) = f(x)/\|f(x)\|_2, \tag{2}$$

where $f$ is a model, $\hat{f}$ is its $L_2$ normalized version, $y$ is label of the input $x$, and $\mathbf{w}_y$ is a classifier vector in $\mathbf{W}_{\text{ETF}}$ for class $y$.

## 4 Approach

When the distribution of the training data is balanced, neural collapse occurs with sufficient training (Papyan et al., 2020). But it does not occur naturally in an imbalanced data distribution (Fang et al., 2021; Yang et al., 2022). Previous studies (Yang et al., 2022; Zhong et al., 2023) induced neural collapse in imbalanced data for the better converged learning using a fixed ETF classifier as its classifier. In offline CL, Yang et al. (2023a) induce neural collapse in a similar way.

However, online CL poses a greater challenge for inducing neural collapse than offline CL, as streamed data are trained only once, while joint training or offline CL allows multi-epoch training. Insufficient training in online CL leads to incomplete convergence towards neural collapse, leading to disappointing anytime inference performance (Koh et al., 2022).

To learn a better converged model without multi-epoch training for online CL, we propose two novel methods, each for the training phase and the inference phase, respectively. In the training phase, we accelerate convergence by proposing *preparatory data training* (Sec. 4.2). In the inference phase, we propose to correct the discrepancy in alignment of classifiers and the features using *residual correction* (Sec. 4.3). We illustrate overview of our approach in Fig. 2.

### 4.1 Inducing Neural Collapse for Online Continual Learning

In offline CL, where existing and novel classes are imbalanced, Yang et al. (2023a) recently show significant accuracy improvement by inducing neural collapse using a fixed ETF classifier. Inspired

---

[1]If the training objective includes a contrastive term between different classes like cross entropy, it could causes incorrect gradients (Yang et al., 2022).

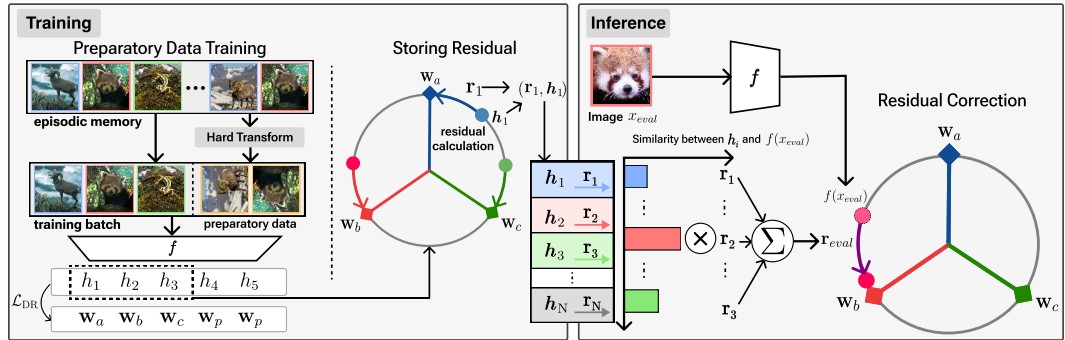

Figure 2: Overview of the proposed method. $\mathbf{w}_i$ denotes the ETF classifier vector for class $i$. $h_a$ denotes the output of the model. The colors of the data denote the class to which the data belongs. Arrow $\mathbf{r}_i$ denotes the residual between the last layer activation $h_i$ and the classifier vector $\mathbf{w}_i$ for class $i$. During training, both memory data and preparatory data are used for replaying, and the residuals between $h_i$ and $\mathbf{w}_i$ are stored in feature-residual memory. During inference, using similarity between $h_i$ and $h$ in feature-residual memory, $\mathbf{r}_{\text{eval}}$ is obtained by a weighted sum of $\mathbf{r}_i$'s. Finally, by adding $\mathbf{r}_{\text{eval}}$, $f(x_{\text{eval}})$ is corrected. Purple arrow indicates 'residual correction' (Sec. 4.3).

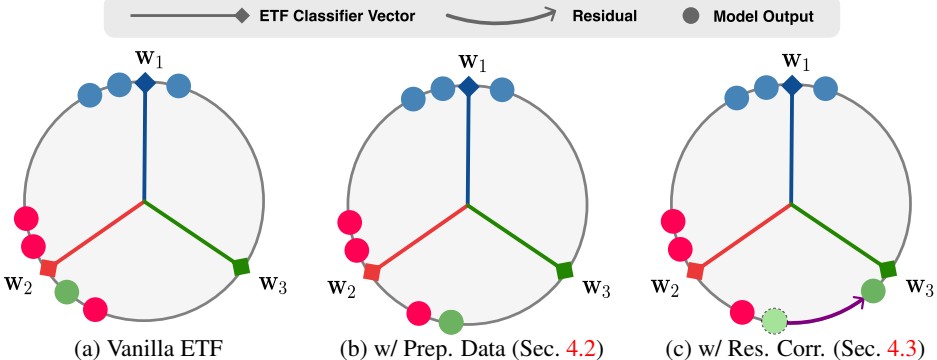

(a) Vanilla ETF      (b) w/ Prep. Data (Sec. 4.2)      (c) w/ Res. Corr. (Sec. 4.3)

Figure 3: ETF structures. In online CL, (a) features of novel classes are biased towards the features of the previous class. (b) By training with preparatory data (**Prep. Data**), we address the bias problem. (c) At inference, features where it does not fully converge to an ETF classifier, we add residuals (**Res. Corr.**) to features that have not yet reached the corresponding classifier vectors, making features aligned with them. Purple arrow indicates 'residual correction' (Sec. 4.3). Colors denote classes.

by this work, we try to induce neural collapse in online CL. Yang et al. (2023a) used the total number of classes $K$ in advance as the number of classifier vectors in the ETF classifier so that $\mathbf{W}_{\text{ETF}} \in R^{d \times K}$, where $d$ is the dimension of the embedding space. However, in CL, it is impossible to know $K$ as it evolves in time.

For realistic, online CL, we first propose to use the maximum possible number of classifier vectors instead of $K$. In a $d$-dimensional output embedding space $f(x) \in \mathbb{R}^d$, the maximum cardinality of a mutually orthogonal vector set is $d$. Since the ETF classifier is defined by a partial orthogonal matrix (Papyan et al., 2020), a maximum of $d$ classifier vectors can be created for an ETF classifier. Therefore, we define the ETF classifier as $\mathbf{W}_{\text{ETF}} \in \mathbb{R}^{d \times d}$, which does not require $K$. Note that we use episodic memory for replay, which is widely used in online CL (Aljundi et al., 2018; Koh et al., 2022; Mai et al., 2021).

## 4.2 PREPARATORY DATA AT TRAINING

Since novel classes arrive continuously in CL, both data from previous tasks ($x_{\text{old}}$) and the current task ($x_{\text{new}}$) coexist. While $\hat{f}(x_{\text{old}})$ are placed closer to their corresponding ETF classifier vectors $w_{\text{new}}$ by training, $\hat{f}(x_{\text{new}})$ are biased towards the cluster of $\hat{f}(x_{\text{old}})$, as we can see in Fig. 4-(a), where $\hat{f}(x_{\text{new}})$ and $\hat{f}(x_{\text{old}})$ is the output of the model for input $x_{\text{new}}$ and $x_{\text{old}}$, respectively. We

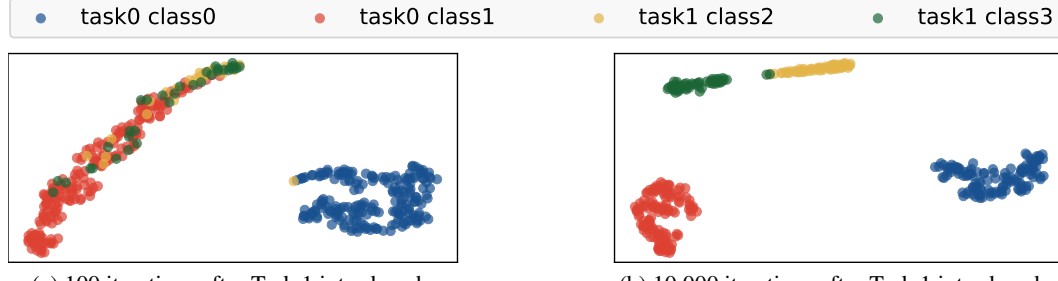

| (a) 100 iterations after Task 1 introduced | (b) 10,000 iterations after Task 1 introduced |

Figure 4: t-SNE visualization of data distribution (class 0 to 3) for 'bias-problem'. (a) Only after 100 iterations of training after task 1 appears, learning is likely insufficient, and we can see that the features of new classes (class 2, 3) are biased towards the feature cluster of the existing class (*i.e.*, class 1). (b) With more training iterations (10,000 iter), the features are well clustered by class.

call this '*bias problem*.' When $\hat{h}_{\text{new}}$ and $\hat{h}_{\text{old}}$ overlap due to bias and are optimized with the same objective function, the training of the new class interferes with the representations of the old classes, perturbing the well-clustered $\hat{h}_{\text{old}}$ (Caccia et al., 2021). This perturbation destroys the ETF structure formed by $\hat{h}_{\text{old}}$ that hinders convergence toward neural collapse.

To accelerate convergence in the ETF structure, we propose to prevent novel classes from being biased towards major classes when introduced, mitigating the bias problem. Specifically, we train a model to avoid making predictions in favor of the existing classes for images that do not belong to them. In particular, we propose preparatory data $x_p$ different from existing classes obtained by transforming the samples in the episodic memory. By training that preparatory data is different from the existing classes, we prevent biased predictions towards the existing classes when a new class arrives.

Specifically, for a set of existing classes $C$ and the set of possible transformations $T$, we randomly select $c \in C$ and $t \in T$, and randomly retrieve a sample from class $c$ from memory and apply the transformation $t$ to obtain *preparatory data* $x_p$. We assign the labels of the unseen classes to the preparatory data by a mapping function $m : C \times T \to C'$ where $C'$ denotes set of unseen classes, *i.e.*, $C' = \{c | c \notin C, \ 1 \le c \le d\}$ and $d$ is the total number of classifier vectors in $\mathbf{W}_{\text{ETF}} \in \mathbb{R}^{d \times d}$. Thus, preparatory data $x_p$ from class $c$ and transformation $t$ is pulled toward the classifier vector $\mathbf{w}_p$, where $p = m(c, t)$. When a new class $c_{\text{new}}$ is added to $C$, we update $m$ by randomly assigning a new mapping $m(c_{\text{new}}, t)$ for $c_{\text{new}}$ and $t \in T$.

We compose the transformation set $T$ as a hard transformation that can modify semantic information (*i.e.*, change the label), used in self-supervised literature (Gidaris et al., 2018; Feng et al., 2019). Specifically, we used rotation by 90, 180, and 270 degrees (Feng et al., 2019; Gidaris et al., 2018) as our hard transformation because the rotation is simple to implement, widely applicable from low- to high-resolution images, and also outperforms other transformations on our empirical evaluations. More detailed analysis of the hard transformation is provided in Appendix Sec. A.2.

Formally, for a supervised training data $(x, y)$, we generate preparatory data $x_p$ (Sec. 4.2), and we write a simple data augmented objective with them as:

$$\mathcal{L}(x, y) = \mathcal{L}_{\text{DR}}(\hat{f}(x), y; \mathbf{w}_y) + \sum_{x_p \in T(x)} \mathcal{L}_{\text{DR}}(\hat{f}(x_p), y; \mathbf{w}_p), \tag{3}$$

where $\mathbf{w}_y$ and $\mathbf{w}_p$ are corresponding classifier vectors.

### 4.3 RESIDUAL CORRECTION AT INFERENCE

Despite the preparatory data training that accelerates convergence towards ETF, in online CL, new samples in a data stream hinder models from reaching the TPT and fully converging to the ETF during the single-epoch training. When model output $f(x)$ has not fully converged to the ETF classifier, model would not perform well at all time, leading low anytime accuracy ($A_{\text{AUC}}$).

To address this issue, we want to correct the residual between the $f(x)$ and the corresponding classifier vector $w_y$ during inference, where $y$ is label of input $x$. However, since the discrepancy between the prediction on the test data $x_{\text{eval}}$ and the ground truth ETF classifier vector is unavailable at training phase. In similar situation, Resmem (Yang et al., 2023b) stores the residual obtained from the training data and used them during inference. Inspired by that, we propose to use the residual obtained from the training process to estimate the residual correction at inference.

To select which of the stored residuals to use during inference, we not only store the residual, but also $f(x)$ to choose the stored residual closest to $f(x_{\text{infer}})$. Therefore, we retain 'feature-residual' pairs in a 'feature-residual memory' $M = \{(\hat{h}_i, \mathbf{r}_i)\}_{i=1}^N$, where $\hat{h}_i = \hat{f}(x_i)$, $\mathbf{r}_i = \mathbf{w}_{y_i} - \hat{f}(x_i)$, where $N$ is the size of feature-residual memory, and $\mathbf{w}_{y_i}$ is the classifier vector for class $y_i$.

During inference, we select the $k$ nearest-neighbor $\hat{h}$'s, $i.e.$, $\{\hat{h}_{n_1}, \hat{h}_{n_1}, \ldots, \hat{h}_{n_k}\}$ from $\{\hat{h}_i\}_{i=1}^N$, since using only the nearest residual for correction may lead to incorrect inference predictions if a wrong residual is selected from a different class. Finally, we calculate the residual-correcting term $\mathbf{r}_{\text{eval}}$ by a weighted average of the corresponding residuals $\{\mathbf{r}_{n_1}, \mathbf{r}_{n_2}, \ldots, \mathbf{r}_{n_k}\}$, with weights $\{s_1, s_2, \ldots, s_k\}$ that are inversely proportional to the distance from $\hat{f}(x_{\text{eval}})$, as:

$$\mathbf{r}_{\text{eval}} = \sum_{i=1}^k s_i \mathbf{r}_i, \quad s_i = \frac{e^{-(\hat{f}(x_{\text{eval}}) - \hat{h}_i)/\tau}}{\sum_{j=1}^k e^{-(\hat{f}(x_{\text{eval}}) - \hat{h}_j)/\tau}}, \tag{4}$$

where $\tau$ is softmax temperature. We add the residual-correcting term $\mathbf{r}_{\text{eval}}$ to the model output $\hat{f}(x_{\text{eval}})$ to obtain the corrected output $\hat{f}(x_{\text{eval}})_{\text{corrected}}$ as:

$$\hat{f}(x_{\text{eval}})_{\text{corrected}} = \hat{f}(x_{\text{eval}}) + \mathbf{r}_{\text{eval}}. \tag{5}$$

Please refer to Appendix Sec. A.6 for detailed comparison between Resmem (Yang et al., 2023b) and our residual addition.

## 5 EXPERIMENTS

### 5.1 EXPERIMENTAL SETUP

We performed experiments on four datasets: CIFAR10, CIFAR100, TinyImageNet, and ImageNet-200. Due to computational resource constraints (Bang et al., 2021; Koh et al., 2022), we used ImageNet-200 by subsampling data for 200 randomly selected classes from ImageNet. For all datasets, our experiments are conducted on both a Disjoint setup (Parisi et al., 2019) and a Gaussian scheduled setup (Shanahan et al., 2021; Wang et al., 2022). We report the average and standard deviation results in three different seeds. For evaluation, to evaluate anytime inference performance, we used the area under the curve accuracy($A_{auc}$) (Koh et al., 2022; Caccia et al., 2022), which measures the area under the accuracy curve. We also used last accuracy($A_{last}$) which measures accuracy at the point when training has been completed for all samples. For detailed information about the experimental setup, refer to Appendix Sec. A.3.

**Baselines.** We compared EARL with the following other baselines: EWC (Kirkpatrick et al., 2017), ER (Rolnick et al., 2019), ER-MIR (Aljundi et al., 2019), REMIND (Hayes et al., 2020), DER++ (Buzzega et al., 2020), SCR (Mai et al., 2021), MEMO (Zhou et al., 2022a), and ODDL (Boschini et al., 2022). For more details on the implementation of these methods, please refer to Sec.A.7. For a comparison with NC-FSCIL (Yang et al., 2023a), which attempts to induce neural collapse in the context of few-shot class incremental learning, see Sec. A.8

**Implementation details.** We use three components to architect our model: a backbone network $g(\cdot)$, a projection MLP $p(\cdot)$, and a fixed ETF classifier $\mathbf{W}_{\text{ETF}}$ (*i.e.*, our model $f$ can be defined as $f(x) = p \circ g(x)$). For the projection layer, we attach an MLP projection layer $p_{\theta_p}$ to the output of the backbone network $g_{\theta_g}$, where $\theta_g$ and $\theta_p$ denote the parameters of the backbone network and the projection layer, respectively (*i.e.*, model $f$ can be defined as $f(x) = p \circ g(x)$), following (Chen et al., 2020; Peng et al., 2022; Yang et al., 2023a). For all methods, we use ResNet-18 (He et al., 2016) as the backbone network.

Following (Koh et al., 2022; Ye & Bors, 2022b), we employ memory-only training, where a random batch is selected from the memory at each iteration. Furthermore, for episodic memory sampling,

| Methods | CIFAR-10 | | | | CIFAR-100 | | | |
|---|---|---|---|---|---|---|---|---|
| | Disjoint | | Gaussian-Scheduled | | Disjoint | | Gaussian-Scheduled | |
| | $A_{\text{AUC}} \uparrow$ | $A_{last} \uparrow$ | $A_{\text{AUC}} \uparrow$ | $A_{last} \uparrow$ | $A_{\text{AUC}} \uparrow$ | $A_{last} \uparrow$ | $A_{\text{AUC}} \uparrow$ | $A_{last} \uparrow$ |
| ER | 75.94±0.86 | 63.56±1.32 | 60.13±0.56 | 64.81±2.70 | 52.95±1.25 | 42.82±0.05 | 41.12±0.56 | 42.74±1.09 |
| DER++ | 74.57±0.89 | 60.80±1.31 | 59.88±0.37 | 64.75±2.29 | 54.51±1.18 | 42.86±0.63 | 43.28±0.57 | 44.60±1.46 |
| ER-MIR | 75.89±1.02 | 61.93±0.93 | 60.39±0.48 | 61.64±3.86 | 52.93±1.44 | 42.47±0.13 | 41.19±0.63 | 42.93± 1.18 |
| SCR | 75.61±0.93 | 56.52±0.52 | 60.62±0.43 | 58.41±2.39 | 41.84±0.74 | 36.00±0.83 | 31.33±0.41 | 32.11±0.39 |
| EWC | 75.25±0.78 | 60.80±2.20 | 59.62±0.31 | 64.24±1.97 | 52.08±0.83 | 41.55±0.85 | 38.22±0.50 | 42.52±0.58 |
| REMIND | 69.55±0.91 | 53.34±1.01 | 58.01±0.72 | 59.27±1.86 | 40.87±0.76 | 36.17±1.83 | 23.40±2.25 | 28.78±1.71 |
| X-DER | 74.34±0.40 | 62.31±2.28 | 57.05±3.76 | 62.89±2.71 | 52.80±1.61 | 43.73±0.86 | 41.94±0.57 | 44.90±1.04 |
| ODDL | 75.03±1.00 | 61.61±3.55 | 65.46±0.46 | 66.19±2.08 | 40.26±0.50 | 41.88±4.52 | 38.82±0.49 | 41.35±1.08 |
| MEMO | 73.21±0.49 | 62.47±3.38 | 59.26±0.90 | 62.01±1.17 | 40.60±1.11 | 39.87±0.46 | 23.41±1.63 | 32.74±2.11 |
| EARL (Ours) | **79.62±0.62** | **67.58±1.51** | **70.56±0.37** | **71.46±1.00** | **57.12±1.22** | **45.15±0.68** | **48.05±0.49** | **46.59±0.35** |

| Methods | TinyImageNet | | | | ImageNet-200 | | | |
|---|---|---|---|---|---|---|---|---|
| | Disjoint | | Gaussian-Scheduled | | Disjoint | | Gaussian-Scheduled | |
| | $A_{\text{AUC}} \uparrow$ | $A_{last} \uparrow$ | $A_{\text{AUC}} \uparrow$ | $A_{last} \uparrow$ | $A_{\text{AUC}} \uparrow$ | $A_{last} \uparrow$ | $A_{\text{AUC}} \uparrow$ | $A_{last} \uparrow$ |
| ER | 37.43±1.05 | 27.47±0.63 | 26.37±0.89 | 25.79±0.44 | 41.51±0.76 | 30.87±0.72 | 32.39±0.36 | 33.09±0.37 |
| DER++ | 38.05±1.07 | 25.41±0.50 | 31.04±0.67 | 27.68±0.77 | 43.20±0.31 | 34.06±0.50 | 35.22±0.26 | 37.88±0.97 |
| ER-MIR | 37.81±1.06 | 26.72±0.86 | 26.22±0.69 | 25.11±1.04 | 38.28±0.38 | 33.12±0.73 | 32.17±0.44 | 33.85±0.93 |
| SCR | 34.65±1.08 | 22.18±0.32 | 25.86±0.94 | 22.54±0.59 | 41.90±0.40 | 28.92±0.40 | 33.24±0.32 | 30.98±0.28 |
| EWC | 37.95±0.93 | 27.50±0.80 | 25.29±0.81 | 26.06±0.52 | 41.84±0.64 | 31.57±0.80 | 30.71±0.27 | 33.33±0.98 |
| REMIND | 28.37±0.13 | 27.68±0.45 | 10.19±0.60 | 14.90±1.49 | 39.25±0.93 | 31.98±0.84 | 30.23±0.62 | 33.98±0.09 |
| X-DER | 35.15±2.12 | 26.67±0.52 | 29.71±0.86 | 28.10±0.50 | 43.21±0.47 | 33.84±0.98 | 36.31±0.17 | 38.31±0.55 |
| MEMO | 27.36±0.61 | 27.57±0.52 | 10.82±1.23 | 18.03±1.36 | 41.55±0.23 | 34.19±1.47 | 32.54±0.39 | 36.11±1.06 |
| EARL (Ours) | **41.77±1.26** | **29.65±0.20** | **35.08±0.70** | **32.49±1.21** | **44.88±0.29** | **34.27±0.55** | **39.14±0.47** | **38.83±0.35** |

Table 1: Comparison of online CL methods on Disjoint and Gaussian Scheduled Setup for CIFAR10, CIFAR100, TinyImageNet and ImageNet-200.

| METHOD | CIFAR10 | | | | CIFAR100 | | | |
|---|---|---|---|---|---|---|---|---|
| | Disjoint | | Gaussian-Scheduled | | Disjoint | | Gaussian-Scheduled | |
| | $A_{\text{AUC}} \uparrow$ | $A_{last} \uparrow$ | $A_{\text{AUC}}$ | $A_{last} \uparrow$ | $A_{\text{AUC}} \uparrow$ | $A_{last} \uparrow$ | $A_{\text{AUC}} \uparrow$ | $A_{last} \uparrow$ |
| EARL (Ours) | **78.61±0.72** | **66.01±2.26** | **69.62±0.19** | **70.91±1.97** | **57.42±1.24** | **44.60±0.65** | **48.19±0.61** | **46.10±0.26** |
| (-) RC | 77.78±0.52 | 65.52±1.63 | 68.37±0.10 | 70.17±1.08 | 56.28±1.40 | 44.29±1.04 | 47.05±0.71 | 46.07±0.52 |
| (-) RC & PDT | 74.77±0.77 | 61.10±3.12 | 65.60±0.25 | 67.39±0.78 | 52.91±1.05 | 41.08±0.60 | 44.34±0.55 | 44.45±0.48 |

Table 2: Ablation Study. RC and PDT refer to preparatory data training (Sec. 4.2) and the residual correction (Sec. 4.3), respectively.

EARL uses the Greedy Balanced Sampling strategy (Prabhu et al., 2020). We describe the details about hyperparameters in Sec. A.4 in Appendix, and the pseudocode of EARL in Sec. A.9 in Appendix for the sake of space.

## 5.2 RESULTS

We first compare the accuracy of the online continual learning methods, including EARL, and summarize the results in Table 1. As shown in the table, EARL outperforms other baselines on all benchmarks, both in disjoint and Gaussian-scheduled setups. In particular, high $A_{\text{AUC}}$ suggests that EARL outperforms other methods for all the time that the data stream is provided to the model.

Furthermore, EARL does not use task boundary information during training, *i.e.*, *task-free*, in contrast to EWC, X-DER, MEMO, and REMIND which use task-boundary information. Nevertheless, EARL outperforms these methods even in the disjoint setup where utilizing task boundary information is advantageous due to abrupt distribution shifts at the task boundaries.

## 5.3 ABLATION STUDY

We conducted an ablation study on the two components of EARL, preparatory data training, and residual correction, and the results are summarized in Table 2. While both components contribute to performance improvement, preparatory data training shows a larger gain in performance. Fig. 5 shows the cosine similarity between the output features of 50 randomly selected samples from the test set of the novel class 4 and the classifier vectors $\mathbf{w}_i$. In baseline (a), the features of the new class 4 are strongly biased towards the classifier vectors of the old classes ($\mathbf{w}_1$, $\mathbf{w}_2$, $\mathbf{w}_3$), rather than $\mathbf{w}_4$. (c) When only residuals are used, there are many samples highly similar to $\mathbf{w}_4$ compared to the baseline.

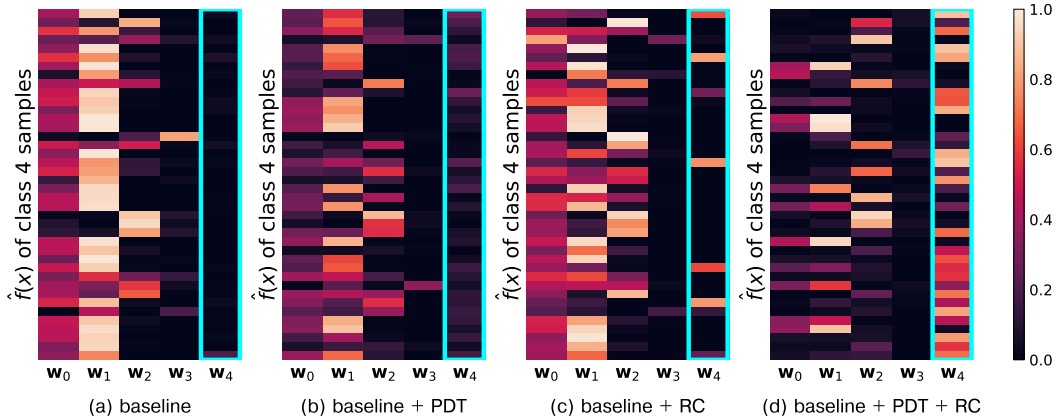

Figure 5: Cosine similarity between the features $\hat{f}(x)$ for class 4 and the ETF classifier vectors $\mathbf{w}_i$ at the $50^{th}$ iteration after the introduction of class 4 in the Gaussian Scheduled CIFAR-10 setup. As we can see in the cyan highlighting box, EARL promotes the convergence of $\hat{f}(x)$ for class 4 toward the ground truth classifier vector $\mathbf{w}_4$.

However, when adding incorrect residuals caused by the bias problem, the similarity between $\mathbf{w}_3$ is higher compared to the baseline. (b) When only preparatory data training is applied, the bias towards $\mathbf{w}_1$ and $\mathbf{w}_2$ classes is significantly reduced compared to the baseline. Fig. 6 shows the effect of preventing bias toward existing classes in the last layer activation of newly arrived classes. (d) Using both residual correction and preparatory data training shows a remarkable alignment with the ground truth classifier $\mathbf{w}_4$. More detailed analysis of ablation results are in Sec. A.10.

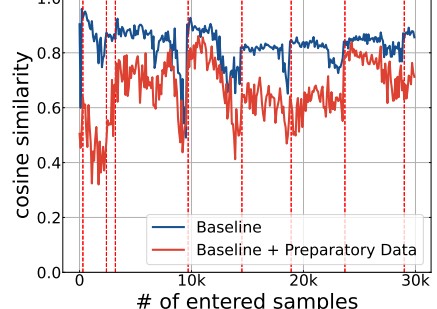

Figure 6: Average of the maximum similarity between the features of the most recently added class's samples and the classifier vectors of the old classes. Baseline is a vanilla ETF model trained only using episodic memory.

Training with preparatory data, our methods outperforms others by $2.7\% \sim 3.3\%$ in $A_{auc}$ and $2.1\%$ $4.4\%$ in $A_{last}$. When combined with residual correction, we observed further gains across all metrics in both disjoint and Gaussian-scheduled setups.

## 6 CONCLUSION

To better learn the online data in a continuous data stream without multiple epoch training, we propose to induce neural collapse, which aligns last layer activations to the corresponding classifier vectors in the representation space. Unlike in offline CL, it is challenging to induce the neural collapse in online CL due to insufficient training epochs and continuously streamed new data. We first observe that the bias of the new class towards existing classes slows the convergence of features toward neural collapse.

To mitigate the bias, we propose synthesizing preparatory data for unseen classes by transforming the samples of existing classes. Using the preparatory data for training , we accelerate the neural collapse in online CL scenario. Additionally, we propose residual correction to resolve the remaining discrepancy toward neural collapse at inference, which arises due to the continuous stream of new data. In our empirical evaluations, the proposed methods outperform state-of-the-art online CL methods in various datasets and setups, especially high performance on anytime inference.

**Limitations and Future Work.** Since our work uses ETF structure, it has an inherent limitation that the number of possible classifier vectors in the ETF classifier is limited by the dimension of the embedding space. Considering lifelong learning, where the number of new classes goes to infinity, it is interesting to explore an idea of dynamically expanding the ETF structure so that the model can continually learn the ever-increasing number of concepts in the real world.

ETHICS STATEMENT

We propose a better learning scheme for the online continual learning for realistic learning scenario. While the authors do not explicitly aim for this, the increasing adoption of deep learning models in real-world contexts with streaming data could potentially raise concerns such as inadvertently introducing biases or discrimination. We note that we are committed to implementing all feasible precautions to avert such consequences, as they are unequivocally contrary to our intentions.

REPRODUCIBILITY STATEMENT

We take reproducibility in deep learning very seriously and highlight some of the contents of the manuscript that might help to reproduce our work. We will definitely release our implementation of the proposed method in Sec. 4, the data splits and the baselines used in our experiments in Sec. 5.

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

## A APPENDIX

### A.1 PROBLEM STATEMENT OF ONLINE CONTINUAL LEARNING

In continual learning (CL), a sequence of tasks $\mathcal{T} = (T_1, T_2, \cdots, )$ is given, where each task $T_i$ is a dataset with training data $D_i = \{(x_1^{(i)}, y_1^{(i)}), (x_2^{(i)}, y_2^{(i)}), \cdots, \}$. Starting from the initial model $\theta_0$, an offline CL algorithm $A_{\text{CL}}$ obtains $\theta_k$, the model for task $k$, from previous model $\theta_{k-1}$ and current task data $D_k$ as $\theta_k = A_{\text{CL}}(\theta_{k-1}, D_k)$. In addition, many CL setups allow the use of episodic memory $M_k$, which is a limited-size subset of training data from previous tasks, *i.e.*, $\theta_k, M_k = A_{\text{CL}}(\theta_{k-1}, M_{k-1}, D_k)$. The objective is to minimize the error of $\theta_k$ on all observed tasks $\{T_i\}_{i=1}^k$.

Unlike offline CL where the whole task data $D_k$ is given as the input, in an online CL, the input is provided as a stream of samples $(x_1^{(k)}, y_1^{(k)}), (x_2^{(k)}, y_2^{(k)}), \cdots$. Thus, an online CL algorithm $A_{\text{OCL}}$ is defined as: $\theta_{k,t}, M_{k,t} = A_{\text{OCL}}\left(\theta_{k,t-1}, M_{k,t-1}, (x_t^{(k)}, y_t^{(k)})\right)$, with the same objective of minimize the error of $\theta_{k,t}$ on $\{T_i\}_{i=1}^k$. Since the model doesn't have access to $(x_1^{(k)}, y_1^{(k)}), \cdots, (x_{t-1}^{(k)}, y_{t-1}^{(k)}) \in D_k$ at time $t$, the multi-epoch training with $D_k$ is not possible in online CL, which some previous works use as a definition of online CL.

### A.2 ANALYSIS ABOUT HARD TRANSFORMATION

Considering various hard transformations such as rotation by 90, 180, and 270 degrees (Feng et al., 2019; Gidaris et al., 2018), patch permutation (Doersch et al., 2015), and small mixing patches using CutMix (Yun et al., 2019), we choose to use rotation 90, 180, and 270 degrees as the hard transform set $T$, because the rotation transformation is simple to implement, widely applicable from low- to high-resolution images, and also outperforms other transformations on our empirical evaluations. On the contrary, patch permutation or small mixing patches could cause discontinuities at the boundary of patches (Kimata et al., 2022; Lee et al., 2022), leading to loss of image features. Performance comparison of hard transformations is in Tab. 3.

| Hard Transformation | Gaussian-Scheduled | | Disjoint | |
|---|---|---|---|---|
| | $A_{\text{AUC}}$ ↑ | $A_{\text{last}}$ ↑ | $A_{\text{AUC}}$ ↑ | $A_{\text{last}}$ ↑ |
| Patch Permutation | 66.37±0.36 | 69.04±1.39 | 75.34±0.88 | 63.73±2.05 |
| Hard Cutmix | 65.26±0.22 | 64.48±1.07 | 72.86±0.13 | 56.56±3.25 |
| Gaussian noise | 64.83±0.23 | 66.97±1.40 | 74.18±0.22 | 63.32±1.52 |
| Rotation | 69.52±0.13 | 70.28±1.77 | 77.86±0.71 | 69.50±1.94 |

Table 3: Comparison of various hard transformations to obtain preparatory data on the Gaussian Scheduled setup in CIFAR-10

### A.3 DETAILS ABOUT EXPERIMENT SETUP

This paper focuses on online class-incremental learning in two types of setups: Disjoint and Gaussian scheduled. Disjoint setup corresponds to a setup of which each class is assigned to a certain task. Each task consists of equal number of classes, randomly mixed with each other. Gaussian scheduled setup, on the other hand, is arranged so that each class follows a Gaussian distribution $\mathcal{N}(\mu, \sigma)$ overlapping one another. Each class manifests a single mode, appearing at different time periods with equal intervals from the preceding and succeeding class. Remarkably, this setup is a boundary-free setup, bearing in mind distribution shifts are made not just at task boundaries in the real world. For our experiments, we regulated $\sigma$ to be 0.1 for comprehensiveness.

### A.4 HYPERPARAMETERS.

For all methods, we use Adam optimizer (Kingma & Ba, 2014) with learning rate (LR) of 0.00003 and constant LR scheduler. For data augmentation, we use RandAugment (Cubuk et al., 2020). For

hyperparameters such as online iteration, memory size, and the number of tasks for each dataset, we follow prior works (Prabhu et al., 2020; Koh et al., 2022; Bang et al., 2021). For the number of online iterations and memory size, we use 1, 3, 3, and 0.25, memory sizes of 500, 2,000, 4,000, and 4,000 for the CIFAR-10, CIFAR-100, TinyImageNet, and ImageNet-200 datasets, respectively. To ensure that our proposed method remains data-agnostic, we use the following hyperparameters for all datasets: $c_p = 4$, $\alpha = 2$, $k = 25$, and $\tau = 0.9$.

To ensure a fair comparison among methods, we use the same batch size across all methods. Note that the number of preparatory data is included in the batch size for our method, *i.e.*, batch size = the number of retrieved samples from memory + the number of preparatory data.

## A.5 Details About Dot-Regression Loss.

In the CE loss, the gradient of the last-layer feature can be divided into a 'pull' term that attracts features of the same class toward their corresponding classifier vector and a 'push' term that pushes them away from the classifier vectors of different classes. This allows all classes to be maximally and equally separated. When using a fixed ETF classifier, since the classifier is already optimally fixed, the pull term, which pulls towards the corresponding classifier vector, always works with the correct gradient (Yang et al., 2022). On the other hand, the push term, which pushes away from the classifier vectors of different classes, may result in incorrect gradients. Hence, a dot-regression (DR) loss has been proposed, which has a gradient similar to the pull term of the CE loss, defined as:

$$\mathcal{L}_{DR}(\hat{f}(x),\ \mathbf{W}_{\text{ETF}}) = \frac{1}{2}\Big(\mathbf{w}_{y_i}\hat{f}(x) - 1\Big)^2, \quad \hat{f}(x) = f(x)/\|f(x)\|, \tag{6}$$

where $h_i$ is last layer feature representation $x_i$, $y_i$ is label of input $x_i$, and $\mathbf{w}_y$ refers to classifier vector in $\mathbf{W}_{\text{ETF}}$ for class $y_i$. Previous research revealed DR loss performs better than CE loss, especially in imbalanced data distributions (Yang et al., 2022; 2023a).

## A.6 Comparison Between Resmem and Residual Addition of Ours

There are several differences between Resmem and our residual correction approach. First, in contrast to the two-stage training-evaluation assumption made in Resmem (Yang et al., 2023b), we took into account that inference queries can be received at any time during training in online CL. Second, due to memory constraints in online CL, access is limited to data stored in episodic memory. Therefore, considering memory constraints, only $n_c$ pairs of $(\hat{h}_x, \mathbf{r})$ per class are stored in feature memory, where $n_c = M/\|\mathbf{C}\|$. $\mathbf{C}$ refers to the set of learned classes, and $M$ refers to the size of episodic memory. Finally, memory storage of discarded $(\hat{h}_x, \mathbf{r})$ pairs can bring about utilization of outdated residuals, as the model is continuously updated. To mitigate the pair outdated problem, we maintained memory as a queue to ensure that $(\hat{h}_x, \mathbf{r})$ pairs from the most recent training batch data were saved.

## A.7 Implementation of Baselines

Most of the baselines took multiepoch training for granted. In order to conduct unbiased comparisons with our proposed method, as explained below, we modified these methods to a streaming manner for online CL implementation and, in response, made some other considerable, necessary modifications.

**REMIND.** Other than adjusting REMIND (Hayes et al., 2020) base initialization to an online streaming manner, we facilitated several transformations. As depicted in Table 4, in online CL, it is not necessary to have layers fixed as much as in offline learning. It is rather beneficial to minimize the frozen layers and have sufficient layers fine-tuned for additional classes to compensate the inadequate number of training passes in training the earlier layers during base initialization. We studied the performance on CIFAR10 and CIFAR100 as a function of number of frozen layers and revealed that six (3 blocks) is the optimum number for ResNet-18. Compared to the original seven layer fixation after an offline base initialization, REMIND performed comparatively well when frozen until the sixth layer in online CL, despite the decrease in the quantity of stored features because of

the enlarged sizes of stored features. With respect to the hyperparameters used in REMIND, such as the size of the codebook and the number of codebooks for product quantization, we performed additional hyperparameter-search experiments with CIFAR100, as shown in Table 5, and carried out the obtained hyperparameter in the remaining datasets.

Lastly, for small-sized datasets, instead of utilizing pretrained ImageNet weight, we increased the base initialization sample proportion from 10 percent to 60 percent of total samples. Acknowledging 10 percent is not enough for lower-level layers to represent highly transferable features, especially in an online CL setting, we enlarged the number of base initialization for CIFAR10, CIFAR100, and TinyImageNet. However, since ImageNet was originally experimented without pretrained weight, for ImageNet-200, we followed the same setting with 10 percent base initialization samples and seven layers frozen after baseinitialization.

| Frozen Layers | CIFAR10 | | | | CIFAR100 | | | |
|---|---|---|---|---|---|---|---|---|
| | Disjoint | | Gaussian-Scheduled | | Disjoint | | Gaussian-Scheduled | |
| | $A_{\text{AUC}} \uparrow$ | $A_{\text{last}} \uparrow$ | $A_{\text{AUC}} \uparrow$ | $A_{\text{last}} \uparrow$ | $A_{\text{AUC}} \uparrow$ | $A_{\text{last}} \uparrow$ | $A_{\text{AUC}} \uparrow$ | $A_{\text{last}} \uparrow$ |
| 4 Layers | 66.63±1.24 | 46.24±0.12 | 54.55±0.71 | 45.56±1.54 | 38.73±0.27 | 30.19±0.66 | 23.76±0.86 | 26.94±0.37 |
| 5 Layers | **70.11±0.66** | 51.01±0.79 | 56.47±0.96 | 52.00±1.94 | 41.87±0.05 | **38.81±0.41** | 24.79±1.73 | 32.12±2.81 |
| 6 Layers | 69.55±0.91 | 47.28±3.92 | **57.15±0.71** | **53.40±0.70** | **41.92±0.06** | 37.64±1.09 | **25.56±1.10** | **34.08±1.02** |
| 7 Layers | 68.59±0.10 | **53.71±2.27** | 55.37±0.60 | 52.53±1.98 | 40.52±0.17 | 37.42±0.81 | 23.94±1.30 | 33.28±1.43 |

Table 4: REMIND performance as a function of number of frozen layers in ResNet-18 with CIFAR10 and CIFAR100. Rather than freezing all the layers except the last layer (7 Layers), continuously updating the last two layers and fixing the rest (6 Layers) shows the best performance due to limitation of online CL.

| ♯ of Codebooks | CIFAR100 | | | | Codebook Size | CIFAR100 | | | |
|---|---|---|---|---|---|---|---|---|---|
| | Disjoint | | Gaussian-Scheduled | | | Disjoint | | Gaussian-Scheduled | |
| | $A_{\text{AUC}} \uparrow$ | $A_{\text{last}} \uparrow$ | $A_{\text{AUC}} \uparrow$ | $A_{\text{last}} \uparrow$ | | $A_{\text{AUC}} \uparrow$ | $A_{\text{last}} \uparrow$ | $A_{\text{AUC}} \uparrow$ | $A_{\text{last}} \uparrow$ |
| 8 | 41.07±0.27 | 36.97±0.27 | 25.17±1.02 | **34.14±1.18** | 256 | **41.92±0.06** | **37.64±1.09** | 25.56±1.10 | **34.08±1.02** |
| 16 | 41.84±0.29 | 36.48±0.34 | 25.37±0.99 | 33.30±0.26 | 512 | 41.48±0.19 | 36.98±0.51 | **25.85±1.39** | 33.01±1.99 |
| 32 | **41.92±0.06** | **37.64±1.09** | **25.56±1.10** | 34.08±1.02 | 1024 | 41.28±0.18 | 36.48±0.34 | 25.13±0.90 | 31.83±0.19 |
| 64 | 40.13±0.15 | 33.50±0.46 | 24.39±0.85 | 29.48±0.04 | 2048 | 41.11±0.21 | 36.10±0.33 | 25.05±0.90 | 31.33±0.11 |

Table 5: REMIND performance as a function of different codebook sizes and number of codebooks with CIFAR100. Original hyperparameters (codebook size: 256, number of codebooks: 32) consistently show the best performance in online CL setting. The same hyperparameters were used uniformly for all datasets.

**MEMO.** MEMO (Zhou et al., 2022a), a block-expanding model, originally retrieves exemplars, which have a comparatively short distance from the corresponding class mean feature, at every task boundary and train them along with current session data though multiepoch training. In online CL, the model is incapable of accessing all the data for the current session, and requires to continuously update current session data to the memory. This influences both the class mean features of samples in the memory and the exemplars to retrieve from the memory. Therefore, this herding algorithm couldn't be followed as it is, as retrieving samples at every iteration with such tactic would be time-consuming and high-computational. Thus, this retrieval method was modified to a simple class-balanced method. In addition, the model was divided into generalized and specialized blocks, according to the REMIND architecture, as we perceived it would be the ideal architecture for online learning.

**ODDL.** ODDL has been primarily evaluated on datasets up to CIFAR100. Handling datasets beyond TinyImageNet, which contain higher complexity than CIFAR dataset, presents challenges in terms of reconstruction using VAEs. To train a VAE from scratch on a large dataset, substantial computational resources and time are required. In prior research as well, due to these challenges, limited improvement in performance existed on large datasets (Boschini et al., 2022). Thus, for our implementation, we did not conduct experiments on large datasets. In addition, out of the many strategies provided by OODL, we chose random selection as the sample selection strategy.

---

**Algorithm 1** Training Phase

---

1: **Input** model $f_\theta$, Memory $\mathcal{M}$, Residual Memory $\mathcal{M}_{\text{RES}}$, Training data stream $\mathcal{D}$, ETF classifier $\mathbf{W}_{\text{ETF}}$, Hard transformation $\mathcal{R}_r$
2: **for** $(x, y) \in \mathcal{D}$ **do**    ▷ Sample arrives from training data stream D
3:    **Update** $M \leftarrow$ ClassBalancedSampler $(\mathcal{M}, (x, y))$    ▷ Update memory
4:    **Sample** $(X, Y) \leftarrow$ RandomRetrieval$(\mathcal{M})$    ▷ Get batch $(X, Y)$ from memory
5:    **Sample** $(X', Y') \leftarrow$ RandomRetrieval$(\mathcal{M})$    ▷ Get batch $(X', Y')$ to make preparatory data
6:    $(X_{\text{p}}, Y_{\text{p}}) \leftarrow (\mathcal{R}_r(X', Y'))$    ▷ Hard transformation for preparatory data training
7:    $\mathbf{r} = \mathbf{W}_{\text{ETF}y} - f_\theta(x_i)$    ▷ Calculate Residual
8:    **Update** $M_{\text{RES}} \leftarrow ((\mathbf{r}, f_\theta(x))$    ▷ Update residual memory
9:    $\mathcal{L}(X, Y) = L_{DR}(\hat{f}_\theta(X), \mathbf{W}_{\text{ETF}_{\mathbf{y}}}) + L_{DR}(\hat{f}_\theta(X_p), \mathbf{W}_{\text{ETF}_{\mathbf{p}}})$    ▷ Calculate dot-regression loss
10:    **Update** $\theta \leftarrow \theta - \mu \cdot \nabla_\theta \mathcal{L}(X, Y)$    ▷ Update model
11: **end for**
12: **Output** $f_\theta$

---

### A.8 Comparison between NC-FSCIL and Baseline

NC-FSCIL (Yang et al., 2023a) is a recently proposed offline CL method that attempts to induce neural collapse in few-shot class-incremental learning (FSCIL), a class-incremental setting with a few novel classes and training data in each incremental session. NC-FSCIL pre-allocates the optimal, fixed ETF classifier to alleviate the misalignment between the classifier vector $\mathbf{w}$ and the features of the last layer $\hat{f}(x)$ for input $x$, incurred by the imbalanced data distribution of the novel classes. With a backbone network and a projection layer, NC-FSCIL freezes the backbone network after training the base session and further fine-tunes the projection layer in the following incremental sessions, supported with a replay memory of the mean features of each learned class in previous sessions. Although our proposed method also utilizes a fixed ETF classifier, our proposed method has an overall disparate framework and mechanism. The comparative experiment between NC-FSCIL and the baseline of our method is shown in Table 6. The baseline pertains to our proposed approach without the incorporation of preparatory data training and residual correction.

Considering that our method is neither offline CL nor a few-shot class incremental learning, in order to endorse a fair comparison with our proposed method, two modifications have been made to NC-FSCIL. The first modification was the number of frozen layers. We adopted the same optimal freezing configuration that achieved the best performance with REMIND, depicted in Table 4. The second adjustment on NC-FSCIL was on the episodic memory. In online learning, the mean features of each class stored in the memory will be continuously altered, due constant updates on the memory. Also, in a non FSCIL setting, there is a vast amount of samples per class in the each dataset. Correspondingly, storing class-mean features in the buffer memory would rather impede the performance, and therefore features rather than class-mean features should be both stored and retrieved for replay. In addition to that, to recompense the increase in necessary memory size for feature storage, product quantization was exploited.

| Methods | CIFAR10 | | | | CIFAR100 | | | |
| | Disjoint | | Gaussian-Scheduled | | Disjoint | | Gaussian-Scheduled | |
| | $A_{\text{AUC}} \uparrow$ | $A_{\text{last}} \uparrow$ | $A_{\text{AUC}} \uparrow$ | $A_{\text{last}} \uparrow$ | $A_{\text{AUC}} \uparrow$ | $A_{\text{last}} \uparrow$ | $A_{\text{AUC}} \uparrow$ | $A_{\text{last}} \uparrow$ |
|---|---|---|---|---|---|---|---|---|
| NC-FSCIL | 68.09±0.69 | 48.05±1.69 | 53.12±0.77 | 46.11±0.36 | 39.54±0.79 | 34.83±1.27 | 30.91±1.46 | 35.73±1.13 |
| Baseline | **75.27±0.77** | **62.10±4.12** | **65.8±0.25** | **67.79±0.78** | **52.91±1.05** | **41.08±0.60** | **33.34±0.55** | **44.45±0.48** |

Table 6: Quantization comparison between NC-FSCIL and baseline of our method on CIFAR10 and CIFAR100 dataset. Despite NC-FSCIL's high performance in FSCIL, its approach is incompetent in a conventional, online CL setting compared to our baseline.

### A.9 Pseudocode for the Our Method

Algorithm 1 and Algorithm 2 provides detailed pseudocode for EARL.

---

**Algorithm 2** Inference Phase

1: **Input** model $f_\theta$, Memory $\mathcal{M}$, Residual Memory $\mathcal{M}_{\text{RES}}$, Testing data $\mathcal{D}$, ETF classifier $\mathbf{W}_{\text{ETF}}$, number of nearest neighbors $k$
2: **for** $(x, y) \in \mathcal{D}$ **do**            ▷ Sample arrives from testing data D
3:      $(\mathbf{r}_i, \hat{h}_i) \leftarrow \mathcal{M}_{\text{RES}}$            ▷ Get residual and features from residual memory
4:      $\hat{h}_{i=0}^{k-1} \leftarrow \text{KNN}\left(f_\theta(x) - \hat{h}_i\right)$            ▷ Calculate k nearest neighbor features
5:      $\mathbf{r} = \sum_{i=0}^{k-1} s_i \mathbf{r}_i$            ▷ Calculate residual-correcting term
6:      $f_\theta(x)_{\text{corrected}} \leftarrow f_\theta(x) + \mathbf{r}$            ▷ Add residual on features
7:      acc = DotRegresssion($f_\theta(x)_{\text{corrected}} * \mathbf{W}_{\text{ETF}}$, y)            ▷ Calculate accuracy
8: **end for**
9: **Output** acc

---

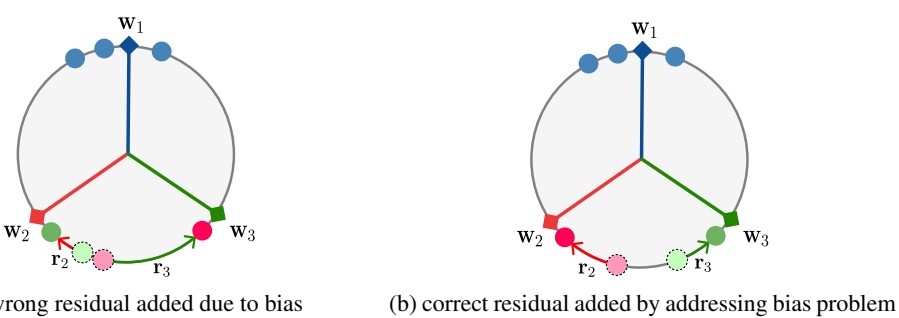

(a) wrong residual added due to bias          (b) correct residual added by addressing bias problem

Figure 7: Without preparatory data training and relying solely on residual correction, incorrect residuals can be added due to bias, which can potentially lead to decreased performance. On the other hand, the combination of preparatory data training and joint training leads to the addition of correct residuals by addressing the bias problem.

## A.10   DETAILED ANALYSIS OF ABLATION RESULTS

Note that exclusively relying on residual correction without the training of preparatory data may result in the addition of incorrect residuals due to bias problem, as illustrated in Fig. 7-(a). The bias in CL causes the features of novel classes and old classes to overlap. When the features of multiple classes are clustered together, as in Fig. 4-(a), residuals of one class can be added to the residual-correcting term of other classes in the cluster, since we select the residuals using $k$ nearest-neighbors of corresponding features. Thus, the residuals of old classes are often added to novel class samples and vice versa, hurting the accuracy of both old and novel classes.

Therefore, the use of preparatory data not only accelerates the convergence of ETF during training, but also promotes accurate residual addition during inference. In conclusion, the combination of residual correction and preparatory data training effectively align the model output with the corresponding ETF classifier, as demonstrated in Fig. 7-(b).

## A.11   EMPIRICAL METRIC FOR NEURAL COLLAPSE

**(NC1) Collapse of Variability**: The last layer feature output of each data point collapses toward the class mean feature of its respective class. In other words, $h_{k,i}$, last layer feature of sample $i$ in class $k$, collapse to $\mu_k = \sum_{i=1}^{n_k} h_{k,i}$ for $\forall k \in [1, K]$ where $n_k$ is the number of samples for class $k$. By considering within-class covariance and between-class covariance

$$\Sigma_W = \frac{1}{K} \sum_{k=1}^{K} \frac{1}{n_k} (\sum_{i=1}^{n_k} (h_{k,i} - \mu_k)), \;\; \Sigma_B = \frac{1}{K} \sum_{k=1}^{K} (\mu_k - \mu_G) \tag{7}$$

where $\mu_G = \sum_{k=1}^{K} \mu_k$, empirical variability can be measured as

$$NC1 := \frac{1}{K} trace(\Sigma_W \Sigma_B^\dagger) \tag{8}$$

**(NC2) Convergence to simplex equiangular tight frame (ETF)**: Class means $\mu_k(k \in [1, K])$ centered by the global mean $\mu_G$ converge to vertices of a simplex ETF structure, i.e., matrix $M = [m_1 \, m_2 \, \cdots \, m_K]$ where $m_k = \frac{\mu_k - \mu_G}{\|\mu_k - \mu_G\|^2}$ satisfies the following equation:

$$\mathbf{M}\mathbf{M}^T = \frac{1}{K-1}(K\boldsymbol{I}_k - \mathbf{1}_K\mathbf{1}_K^T) \tag{9}$$

The degree of convergence can be measured using:

$$\frac{\mathbf{M}\mathbf{M}^T}{\|\mathbf{M}\mathbf{M}^T\|_F} - \frac{1}{\sqrt{K-1}}(\boldsymbol{I}_K - \frac{1}{K}\mathbf{1}_K\mathbf{1}_K^T) \tag{10}$$

**(NC3) Convergence to self-duality**: Classifier $\mathbf{W}$ converges to the simplex ETF $\mathbf{M}$ formed by recentered feature mean, and during this convergence, the classifier vector $w_k$ aligns with their corresponding feature mean $m_k$ where $w_k$ means classifier weight for class $k$, $k \in [1, K]$.

$$\frac{\mathbf{M}}{\|\mathbf{M}\|_F} = \frac{\mathbf{W}}{\|\mathbf{W}\|_F} \tag{11}$$

Duality can be measured by measuring:

$$\frac{\mathbf{W}\mathbf{M}^T}{\|\mathbf{W}\mathbf{M}^T\|_F} - \frac{1}{\sqrt{K-1}}(\boldsymbol{I}_K - \frac{1}{K}\mathbf{1}_K\mathbf{1}_K^T) \tag{12}$$

