# OpenReview forum: "Learning Equi-angular Representations for Online Continual Learning"
_ICLR.cc/2024/Conference — ICLR 2024 Conference Withdrawn Submission_

### Official Review · Reviewer_ubjn · 2023-10-31

**Soundness:** 2 fair
**Presentation:** 2 fair
**Contribution:** 2 fair
**Rating:** 3
**Confidence:** 5

**Summary:**

This paper proposes an "online continual learning" approach, a challenging variant of lifelong learning. Empirical evaluation demonstrates improvements by a noticeable margin over the existing continual learning baselines.

**Strengths:**

1. The paper is well-written and handles a challenging variant of continual learning, online continual learning.
2. Empirical evaluations on various datasets demonstrates the efficacy of the proposed approach in mitigating catastrophic forgetting over existing continual learning baselines.

**Weaknesses:**

I am mainly concerned about empirical evaluation and scalability of the proposed method.

1. This method employs memory only training, however, it might create negative impact in the performances of the existing continual learning baselines. Therefore, I would suggest the authors to train the baselines following the steps mentioned in the respecting proposed baselines, like REMIND, ER-MIR, EWC, DER++.

2. The proposed method uses KNN during inference which could be a time consuming process with higher value of K. Therefore, I believe it is crucial to compare the training time and the inference time of the existing CL baselines with the proposed method by varying K in KNN. Also, report the performance of the proposed method with using $K=1$ in KNN during inference.

3. What would be the performance of the existing CL baselines, if you also include the preparatory data to train the existing CL baselines?

4. What would be the performance of REMIND if the authors use the exact configuration for the pretrained feature extractor as mentioned in REMIND paper?

5. This paper does not compare with various existing CL baselines such as GDumb[1], CLS-ER[2].

[1] Prabhu, Ameya, Philip HS Torr, and Puneet K. Dokania. "Gdumb: A simple approach that questions our progress in continual learning." In Computer Vision–ECCV 2020: 16th European Conference, Glasgow, UK, August 23–28, 2020, Proceedings, Part II 16, pp. 524-540. Springer International Publishing, 2020.

[2] Arani, Elahe, Fahad Sarfraz, and Bahram Zonooz. "Learning fast, learning slow: A general continual learning method based on complementary learning system." arXiv preprint arXiv:2201.12604 (2022).

**Questions:**

Refer to the weaknesses section.

---

### Official Review · Reviewer_Dzox · 2023-10-31

**Soundness:** 2 fair
**Presentation:** 2 fair
**Contribution:** 2 fair
**Rating:** 5
**Confidence:** 5

**Summary:**

The paper introduces a method to enhance plasticity in Continual Learning (CL) by treating CL as an imbalanced problem. Drawing inspiration from the Neural Collapse (NC) phenomenon, the authors suggest a three-component approach. First, they aim to induce Neural Collapse using a fixed regular ETF classifier. Second, they employ preparatory data training with rotated images on pre-allocated free ETF prototypes. Lastly, they design a residual predictor to calculate the necessary shift in a feature to align it closer to a fixed ETF, which in turn aids in the induction of NC.

**Strengths:**

The manuscript in general is clearly articulated and addresses a relevant problem. The combination of the proposed three-component approach appears to be the primary contribution of the paper. Furthermore, inducing neural collapse in continual learning prior to reaching the TPT phase while minimizing perturbation of the old class poses a challenging aspect.

**Weaknesses:**

The introduction section could benefit from further refinement to enhance clarity. It would be helpful if the central phenomenon driving the paper was elaborated upon more explicitly. For example, the relationship between Continual Learning as an inherently imbalanced problem and the role of Neural Collapse in addressing unbalanced data could be made clearer, especially in the context of new classes being biased towards the features of older classes. Specifically, the manuscript mentions: “the phenomenon in which the features of the new class become biased towards the features of the existing classes hinders fast convergence...” It's commonly understood that this phenomenon might be observed even in the absence of a fixed simplex ETF classifier (and such forcing often results in forgetting). Could the authors clarify the unique aspects or potential novelty of their proposed solution in this context?

The reviewer appreciates the comprehensive coverage in the related work section. However, it might be more beneficial if the content closely aligns with the problem addressed in the paper. While the section provides a detailed list of papers on continual learning, drawing clearer connections to the main topic could enhance its relevance. For instance, the mention of 'task identity' appears to be introduced without prior context or its relationship to the paper's main theme. It might be helpful to either establish a connection with 'collapse' or consider refraining from introducing the 'task ID' without clear relevance.

The paper might benefit from considering additional related works to clarify its focus. Specifically, references regarding the fixed simplex ETF in continual learning could be further explored. For instance, a study introducing simplex fixed classifiers in Class Incremental Learning [2] found no significant improvement with respect to using  trainable classifiers. This has been further confirmed more recently in [8], and [3]. This result is also consistent with observations in [5, 4, 6, 7] regarding standard classification (i.e., no catastrophic forgetting). Some clarification in this regard on the key aspects that contribute to improving performance should be considered.

The section 4.1 asserting that Fixed Simplex ETF induces neural collapse may benefit from further elaboration, supplemented by relevant references or clear numerical evidence. Fixed Simplex ETF classifiers in unbalanced data regime are shown to prevent class prototypes from collapsing onto each other [9] but it doesn't necessarily lead to the neural collapse phenomenon as traditionally understood. The inspiration drawn from this observation should be more thoroughly verified.

The statement in the final paragraph of section 4.1 should be improved for clarity. The chain of implications is not clear: “a maximum of d classifier vectors can be created for an ETF classifier”. “Therefore we define the ETF that does not require K”. To the best understanding of the reviewer In a simplex ETF the number of class K and d seems to be related by K<=d+1. What does exactly mean that the ETF is not requiring K? Moreover is the simplex ETF a regular simplex o a simplex? It seems that near orthogonality.

The method of augmenting existing classes by rotating them into new classes is interesting. Further motivation might improve the novelty of this approach. It would be valuable for example to clarify the distinctions between this approach and the addition of a new dataset with its own labels.  Some further details about the relationship between rotation and neural collapse should be given. Furthermore, the section discussing the incremental preallocation of the simplex ETF would benefit from clearer referencing regarding its distinctions from [2].

The paper about residual correction should be discussed with respect to [1]. Additionally, an explanation for why this is expected to outperform the DNN model's direct predictions and the benefits of decoupling the contribution would be helpful.

Overview: The proposed method encompasses several distinct components, giving it a somewhat fragmented impression. Additionally, some of the assertions made in the paper might benefit from more comprehensive evidence or explanations for clearer validation. It would also be advantageous to have a more thorough revision of the literature, ensuring that foundational works are adequately discussed and integrated.

References

[1] Yu, Lu, et al. "Semantic drift compensation for class-incremental learning." Proceedings of the IEEE/CVF conference on computer vision and pattern recognition. 2020.

[2] Pernici, Federico, et al. "Class-incremental learning with pre-allocated fixed classifiers." 2020 25th International Conference on Pattern Recognition (ICPR). IEEE, 2021.

[3] Boschini, Matteo, et al. "Class-incremental continual learning into the extended der-verse." IEEE Transactions on Pattern Analysis and Machine Intelligence 45.5 (2022): 5497-5512.

[4] Pernici, Federico, et al. "Fix your features: Stationary and maximally discriminative embeddings using regular polytope (fixed classifier) networks." arXiv preprint arXiv:1902.10441 (2019).

[5] Hoffer, Elad, et al. "Fix your classifier: the marginal value of training the last weight layer." International Conference on Learning Representations. 2018.

[6] Pernici, Federico, et al. "Regular polytope networks." IEEE Transactions on Neural Networks and Learning Systems 33.9 (2021): 4373-4387.

[7] Zhu, Zhihui, et al. "A geometric analysis of neural collapse with unconstrained features." Advances in Neural Information Processing Systems 34 (2021): 29820-29834.

[8] Yang, Yibo, et al. "Neural Collapse Inspired Feature-Classifier Alignment for Few-Shot Class-Incremental Learning." The Eleventh International Conference on Learning Representations. 2023.

[9] Yang, Yibo, et al. "Inducing Neural Collapse in Imbalanced Learning: Do We Really Need a Learnable Classifier at the End of Deep Neural Network?." Advances in Neural Information Processing Systems 35 (2022): 37991-38002.

**Questions:**

The reviewer prefers presenting both questions and weaknesses alongside their respective suggestions as it contributes to a better understanding and association of the issues.

---

### Official Review · Reviewer_2VjM · 2023-11-01

**Soundness:** 3 good
**Presentation:** 3 good
**Contribution:** 3 good
**Rating:** 6
**Confidence:** 3

**Summary:**

In this paper, the authors show that in the online continual setting, the prediction of newly arrived samples on new classes can be biased toward old classes because of the insufficient training on new samples. To resolve this issue, the authors induce neural collapse to promote  fitting the streamed data by using preparatory data training and storing the residual information. In the experiment, the proposed algorithm outperforms other baselines in various experiment settings.

**Strengths:**

1. Inducing the neural collapse to accelerate fitting newly arrived data is a novel approach in CIL. In the experiment, the authors show the effectiveness of using ETF classifier with preparatory data training and storing the residual information by showing the degradation of cosine similarity between the features, and also increased the performance.

**Weaknesses:**

1. Though it is hard to achieve remarkable performance in online continual learning scenario with large-scale datasets (e.g. ImageNet-1K), to strengthen the results, it would be better to carry out the large-scale dataset experiment with the proposed algorithm.

**Questions:**

2. Why using preparatory data training can accelerate the convergence? Does the biased samples slow down the convergence speed? I think it would be better to give more detailed explanation on using the preparatory data training.

---

### Official Review · Reviewer_Ff4L · 2023-11-04

**Soundness:** 2 fair
**Presentation:** 3 good
**Contribution:** 2 fair
**Rating:** 3
**Confidence:** 4

**Summary:**

This paper proposes a method called EARL, which seeks to accelerate neural collapse in online continual learning. During training, EARL uses preparatory data to prevent new classes from being biased towards old classes. During inference, EARL applies a residual correction to compensate for not having fully reached neural collapse, in order to improve its accuracy.

**Strengths:**

- The paper was an interesting read.
- The paper does a good job of introducing the concepts of neural collapse and equiangular tight frames.

**Weaknesses:**

- I think there might not be sufficient novelty for publication at a top venue like ICLR.
- I am not sure whether the construction of preparatory data is sound. Why would the rotation of an image change its class?
- The comparison with other methods is probably not completely fair, since EARL uses up more memory resources (due the storage of feature-residual pairs).
- Moreover, EARL is more expensive than a simple classifier at inference time, due to it performing residual correction. I don't know if using residual correction is a good tradeoff, since it only increases the accuracy marginally (as we can see in Table 2).
- Some of the stronger online continual learning baselines are missing (e.g., the asymmetric cross-entropy [1]).
- I am not sure whether EARL would be applicable in data domains for which data augmentation is not as straightforward as with images.

[1] Caccia, L., Aljundi, R., Asadi, N., Tuytelaars, T., Pineau, J., & Belilovsky, E. (2021, October). New Insights on Reducing Abrupt Representation Change in Online Continual Learning. In International Conference on Learning Representations.

**Questions:**

- How does the rotation transformation change the class of an image?
- What is the computational cost of the extra components (i.e., the construction of preparatory data, and the residual correction)?
- How large is the memory that contains feature-residual pairs? Do you store pairs from the entire duration of training, or only from the $k$ most recent steps?
- What is the "number of online iterations" mentioned in the appendix?
- You write that you have used the same batch size for all compared methods, but I could not find the actual value. Did you only test with one value, or multiple?